# Evaluation of candidate reference genes for gene expression analysis in the brassica leaf beetle, *Phaedon brassicae* (Coleoptera: Chrysomelidae)

**Long Ma**[1], **Ting Jiang**[2], **Xiangya Liu**[2], **Haijun Xiao**[2], **Yingchuan Peng**[2], **Wanna Zhang**[2]*

**1** College of Life Sciences, Jiangxi Science & Technology Normal University, Nanchang, China, **2** Institute of Entomology, Jiangxi Agricultural University, Nanchang, China

* zhangwanna880210@yeah.net

**Data Availability Statement:** All relevant data are within the paper and its Supporting information files.

## Abstract

The brassica leaf beetle *Phaedon brassicae* is a notorious defoliator of cruciferous vegetables. However, few molecular studies of this pest have been conducted due to limited sequence data. Recently, RNA sequencing has offered a powerful platform to generate numerous transcriptomic data, which require RT-qPCR to validate target gene expression. The selection of reliable reference genes to normalize RT-qPCR data is a prerequisite for gene expression analysis. In the present study, the expression stabilities of eight candidate reference genes under biotic conditions (development stages and various tissues) and abiotic perturbations (thermal stress and pesticide exposure) were evaluated using four different statistical algorithms. The optimal suites of reference genes were recommended for the respective experimental conditions. For tissue expression analysis, *RPL32* and *EF-1α* were recommended as the suitable reference genes. *RPL19* and *TBP* were the optimal reference genes across different developmental stages. *RPL32* and *TBP* were identified as the most suitable references for thermal stress. Furthermore, *RPL32* and *RPL19* were ranked as the best references for insecticide exposure. This work provides a systematic exploration of the optimal reference genes for the respective experimental conditions, and our findings would facilitate molecular studies of *P. brassicae*.

## Introduction

The brassica leaf beetle, *Phaedon brassicae* Baly, is a notorious defoliator of crucifers and is widely distributed in East and South Asia with high fecundity [1, 2]. This beetle escapes from predators by dropping from host plants, and interestingly, the larvae exhibit less frequent dropping behavior than adults in response to attacks [3]. In the Yangtze River Valley, there are two distinct infestation peaks in the field: the single spring generation and the two generations in autumn, which undergo aestivating and hibernating imaginal diapause in soil, separately [1, 4, 5]. This beetle is a typical short-day species in which low temperature enhances the

**Funding:** This research was supported by National Natural Science Foundation of China (31960543, 32060642), Natural Science Foundation of Jiangxi Province (20202BABL203046, 20202ACBL205004), and Science and Technology Program of Department of Education of Jiangxi Province (GJJ201106).

**Competing interests:** The authors have declared that no competing interests exist.

induction of its winter diapause, while high temperature suppresses the incidence of its summer diapause [1, 5]. In the last decades, this beetle has become a secondary chewing pest of brassicaceous vegetables in China, and recently, it has occurred frequently and caused potential threats to vegetable products. However, the application of insecticides is not always effective against *P. brassicae* due to its complex life history and high fecundity [4], and previous studies have explored a cadherin-based peptide as an enhancer for Cry3Aa-based products in controlling *P. brassicae* [2, 6]. To identify novel target genes for controlling *P. brassicae*, the accurate quantification of gene expression under different conditions is indispensable.

Gene transcription patterns in different tissues and developmental stages provide deep insights into their biological functions [7]. Taking advantage of high-throughput transcriptome sequencing, transcriptome analysis and data mining have become efficient in screening differentially expressed genes and quantifying the expression abundance of their transcripts. Real-time quantitative PCR (RT-qPCR) has become a powerful tool for validating gene expression profiles owing to its accuracy, specificity, sensitivity, dynamic range, and reproducibility [8, 9]. However, several factors, such as RNA purity and integrity, reverse transcription and PCR efficiency, and pipetting errors, can affect the accuracy of RT-qPCR [10]. In RT-qPCR, a common practice to calibrate target expression is to measure the expression of an internal control, namely reference gene, synchronously in the same sample [8]. Generally, reference genes are housekeeping genes that are constitutively expressed to maintain basic cellular function, and the selection of suitable reference genes has become a necessary step prior to RT-qPCR. To accurately determine of target gene expression and eliminate the technical variation among the tested samples, one or several reference genes with stable expression are required as internal controls to normalize the data and to make accurate comparisons among experimental conditions. According to the reference genes documented in entomological research, the most commonly used genes are *actin* (*ACT*), *tubulin (TUB)*, *TATA-Box binding protein* (*TBP*), *glyceraldehyde-3-phosphate dehydrogenase* (*GAPDH*), *elongation factor 1-alpha* (*EF-1α*), *ribosomal proteins* (*RPs*), and *18S ribosomal RNA* (*18S*) [11–13]. Ideally, reference genes should exhibit constant expression with respect to different developmental stages, tissues, or treatment conditions and do not co-regulate with the target gene. However, the transcription levels of such reference genes are not always stable under various biotic conditions (e.g. different tissues or developmental stages) or abiotic conditions (e.g. pesticide exposure or thermal stress), possibly leading to inconsistent results [14–16].

To date, a large body of research has suggested that there is no 'universal' reference gene applicable for various experimental conditions and all tissue types, even within the same insect species [10, 15, 17]. For instance, in the predatory lady beetle *Hippodamia convergens*, *28S*, *EF1A*, and *CypA* were the best reference genes across different developmental stages, while *GAPDH*, *CypA*, and *28S* were the most stable in different tissues; *GAPDH* and *CypA* were the most stable under photoperiod conditions [15]. Similarly, in *Aphidius gifuensis*, *Colaphellus bowringi*, *Mythimna separata* and *Harmonia axyridis*, the optimal set of reference genes significantly varies with developmental stage, gender, and diet [12, 18–20]. Taken together, these studies demonstrated that the selection of reliable reference genes under specific experimental conditions was pivotal before normalizing target gene expression.

The objective of this study was to discern the solidly expressed reference genes in *P. brassicae* across different developmental stages, in various tissues and in response to abiotic perturbations (thermal stress and pesticide exposure) for RT-qPCR analysis. For this purpose, four different analytical tools (geNorm, NormFinder, BestKeeper, and the comparative ΔCt algorithm) were used to assess the stability of candidate reference genes. These results were conclusively integrated in RefFinder to provide an overall ranking of the candidate reference genes. A target gene was selected to verify our findings. As a result, optimal sets of reference genes were

recommended for the respective experimental conditions. To our knowledge, this is the first report of a comprehensive evaluation of reference genes in *P. brassicae*, and our results would facilitate the future research on functional studies of *P. brassicae*.

## Materials and methods

### Insect rearing

Individuals of *P. brassicae* were seized from a natural population in the experimental radish plantation of Jiangxi Agricultural University (Nanchang, Jiangxi, China). This strain has been reared for six generations without exposure to chemical insecticides. Laboratory rearing was conducted on radish (*Raphanus sativus* var. *longipinnatus*) leaves in transparent plastic containers (7.5 cm wide at base and 15.0 cm deep) under conditions of 25 ± 1˚C, with relative humidity at 70 ± 10%, and a 12:12 h light:dark photoperiod. Fresh leaves were provided daily.

### Experimental treatment and sample collection

For the gene expression analysis, the expression of candidate reference genes was tested in different developmental stages, tissues, and treatments to evaluate the stability of the candidate genes. To investigate the developmental expressions, specimens were sampled from eggs, larvae (first, second, third and fourth instar), pupae, and adults. For tissue expression analysis, tissues (including head, cuticula, fat body and gut) were dissected from the third-instar larvae in ice-cold phosphate-buffered saline (PBS). For the insecticide exposure experiment, the second-instar larvae were treated with sublethal doses of acetamiprid (2 μg/ml), dinotefuran (5 μg/ml), and abamectin (0.18 μg/ml) for 48 h, respectively, while an equal dose of PBS was applied to individuals set as controls. Briefly, the leaf discs of radish (2 cm diameter) were dipped in the solution with insecticide for 10 s and then left to air-dry at room temperature. Each piece of dipped leaf discs was placed in a Petri dish, into which four larvae were transferred. To examine the influence of temperature, the third-instar larvae were subjected to cold shock (10˚C), heat shock (40˚C), and control condition (25˚C) for 2 h and 4 h, respectively. Each experiment contained at least 24 individuals, and was performed in three biological replicates. All samples were frozen immediately in liquid nitrogen and stored at -80˚C until RNA isolation.

### RNA isolation and cDNA synthesis

Total RNA was isolated using the Trizol reagent (Thermo Scientific, China) following the manufacturer's protocol. The RNA quality was examined by 1% agarose gel electrophoresis, and its quantity was evaluated with an Agilent 2100 Bioanalyzer (Agilent Technologies). RNA samples with a value of OD 260/280 ratio between 1.8 and 2.0 were further applied into cDNA synthesis. After the removal of residual genomic DNA with DNase I (Promega, Madison, USA), 1 μg of purified total RNA was applied to the first-strand cDNA synthesis using a Fast Quant RT kit (Tiangen, Beijing, China).

### Identification of candidate reference genes

To identify the stably expressed genes, eight candidate reference genes, namely *Actin1*, *Actin2*, *EF-1α*, *GAPDH*, *α-Tub*, *RPL19*, *RPL32*, and *TBP*, were selected from the *P. brassicae* RNA-seq transcriptome database (S1 Table). Prior to RT-qPCR detection, the open reading frames of these genes were confirmed by PCR amplification using specific primers. Other primer pairs

used for RT-qPCR were designed using Primer Express software 3.0.1 (Applied Biosystems). Furthermore, primer specificity was screened by 1.5% agarose electrophoresis after PCR amplification.

## Quantitative real-time PCR analysis

RT-qPCR was performed on a CFX96 Touch Real-time PCR detection system (Bio-Rad) using SYBR Green SuperReal PreMix Plus (Tiangen, Beijing, China). Amplifications were carried out under the following conditions, initial denaturation at 95˚C for 10 min followed by 40 cycles of 5 s at 95˚C and for 30 s at 60˚C, followed by a melting curve stage (60 to 95˚C) to confirm gene-specific amplification. After 5-fold dilution of cDNA template, 2 μl cDNA sample was incorporated in RT-qPCR reaction for a total volume of 25 μl. Each reaction was carried out in triplicate and the average cycle threshold (Ct) values of triplicates were calculated. Meanwhile, negative control without template was performed. A standard curve for each primer pair was constructed with serial dilutions of cDNA samples, and the corresponding amplification efficiency for each candidate gene was calculated following the equation: $E = (10^{[-1/\text{slope}]} - 1) \times 100$.

## Stability analysis of reference gene expression

After RT-qPCR measurement, four algorithms, namely geNorm, NormFinder, BestKeeper, and the comparative ΔCt method, were employed to evaluate the stability of each candidate reference gene. In brief, the geNorm algorithm provides its ranking based on the mean pairwise variation (V-value) between all the tested genes to calculate the expression stability value (M) of genes, and a lower M-value indicates a higher stability [8]. In addition, geNorm was utilized to define the optimal number of reference genes credible for normalization, wherein a V-value within a threshold of 0.15 was set to determine whether additional reference genes were necessary. NormFinder, an Excel-based applet, ranks the gene expression stability based on the evaluation of their intra- and inter-group variation and a separate analysis of the sample subgroups in expression [21]. BestKeeper evaluates expression stability based on an index obtained by calculating the Ct set standard deviation (SD) and coefficient of variance (CV) [22]. Finally, a comprehensive analysis tool RefFinder (https://www.heartcure.com.au/reffinder/#) was used to integrate the results of the four different analytical methods and assess the rank of the reference genes based on their geometric mean [23].

## Validation of the recommended reference genes

In insects, small heat shock proteins (sHSPs) function as molecular chaperones to protect cells from harsh conditions, prevent irreversible protein aggregation and become active in response to thermal stress [24]. To examine the reliability of the selected reference genes, the expression levels of *sHSP20.0* (accession number MW538931), a target gene, were examined during exposure of *P. brassicae* to different thermal conditions (10, 25 and 40˚C) for 2 h. After RT-qPCR, the optimal combination of reference genes (*RPL32* and *TBP*), the optimal reference gene (*RPL32*), and the least stable gene (*GAPDH*) were selected for RT-qPCR normalization of the target gene, respectively. It is noteworthy that the normalization against two reference genes was performed using the geometric mean of the normalization factors. The relative expression of *sHSP20.0* in each sample was calculated using the $2^{-\Delta\Delta Ct}$ method [25].

**Table 1. Primers for candidate reference genes used in RT-qPCR analyses.**

| Genes | Accession No. | Primer sequence (5′-3′) | Product length (bp) | Primer efficiency (%E) | Regression coefficient (R$^2$) | Linear regression |
|---|---|---|---|---|---|---|
| *Actin1* | MW509776 | F: TTCCAATTGCTGGTCGAAAC | 204 | 98.75 | 0.9985 | y = -3.3522x+25.024 |
| | | R: AATTCGAGCCGTCGTACCTT | | | | |
| *Actin2* | MW509777 | F: TGTCGTAGTGGATTCGGGAG | 125 | 102.24 | 0.999 | y = -3.2694x+26.274 |
| | | R: ACTTGATGAGGTACCGGGTG | | | | |
| *EF-1α* | MW509779 | F: TAGGTCGTGTGGAAACTGGTG | 167 | 96.76 | 0.9996 | y = -3.4021x+19.701 |
| | | R: TTCCTTGACGGAGACGTTCTT | | | | |
| *GAPDH* | MW509780 | F: CTCTTGTCGGCAAACTCACC | 194 | 104.50 | 0.9994 | y = -3.2185x+22.762 |
| | | R: GATGAAATCGGACGAGACGA | | | | |
| *α-TUB* | MW509778 | F: TGGACAGGATCAGGAAGCTC | 144 | 96.16 | 0.9991 | y = -3.4176x+20.745 |
| | | R: GCTTCGACTTCTTGCCGTAG | | | | |
| *RPL19* | MW509781 | F: GCATTGTGGGTTTGGAAAGA | 157 | 103.12 | 0.9999 | y = -3.2493x+21.884 |
| | | R: CTTCATGTACAGGGCGTGGT | | | | |
| *RPL32* | MW509782 | F: ACTGGCGTAAACCGAAAGGT | 178 | 99.03 | 0.9998 | y = -3.3455x+19.261 |
| | | R: CGGTTCTGCATGAGAAGGAC | | | | |
| *TBP* | MW509783 | F: GCAAGCAGCAAGAAGGTTTG | 207 | 106.75 | 0.9952 | y = -3.1701x+26.167 |
| | | R: GGGTGGCTTTTGGACTTTTC | | | | |

## Results

### Determination of primer specificity and efficiency

For each candidate reference gene, a single amplicon was detected by agarose gel electrophoresis, ranging from 125 to 207 bp (S1 Fig). Consistent with this result, a single peak was observed in the melting curve analysis (S2 Fig). Furthermore, the amplification efficiency of eight candidate reference genes ranged from 96.16 to 106.75%, with correlation coefficients (R$^2$) of more than 99% (Table 1; S3 Fig).

### Ct values of candidate reference genes under different experimental conditions

The mean Ct values for the eight candidate reference genes ranged from 18 to 27 under the given experimental conditions (Fig 1). For different developmental stages, *EF-1α* exhibited the highest expression, followed by *RPL32*, *RPL19* and *α-TUB*. Based on the tissue expression profiles, the most expressed reference gene was *EF-1α*, followed by *RPL19* and *α-TUB*. According to their integrative performance in different treatments, *RPL19* exhibited the lowest variation (below 2 cycles) in expression, and *EF-1α* and *Actin2* showed the highest and lowest transcription levels, respectively.

### Stability analysis of reference genes under different experimental conditions

In regards to the development expression analysis, *RPL19* and *TBP* were ranked as the top two most stably expressed genes by NormFinder and geNorm, and the top candidates were *EF-1α* and *TBP* according to Bestkeeper and ΔCt method, respectively (Table 2). For tissue expression analysis based on Bestkeeper and ΔCt method, the top three candidates were *RPL19*, *RPL32*, and *EF-1α*. Besides, both NormFinder and geNorm recommended *EF-1α* as the most stably expressed gene. Under different temperature conditions, NormFinder and geNorm

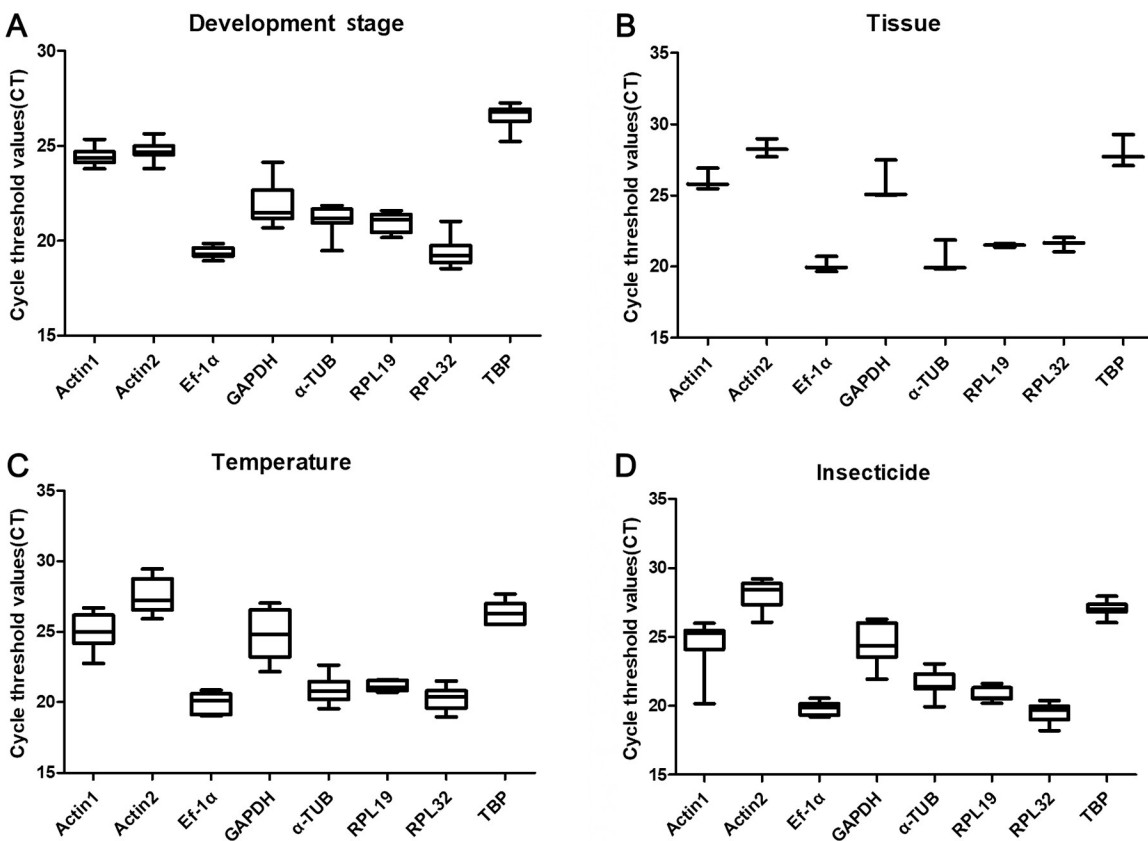

**Fig 1. The cycle threshold (Ct) values of eight reference genes under different treatment conditions.** The vertical bars represent the standard deviation.

ranked *RPL32*, *TBP*, and *α-TUB* as the top three reference genes. Besides, both Bestkeeper and ΔCt method identified *RPL19* as the most stable candidate. In the pesticide exposure experiment, the overall order based on NormFinder from the most stable to the least stable reference gene was: *RPL32*, *α-TUB*, *RPL19*, *Actin2*, *TBP*, *EF-1α*, *GAPDH*, *Actin1*. In addition, Bestkeeper identified *TBP* as the most stable one, and *RPL19* showed the optimal stability based on the ΔCt method.

Integrating the evaluation of four programs, a comprehensive ranking of candidate reference genes was determined by RefFinder (Fig 2; Table 2). The results indicated that *RPL19* and *TBP* were considered as the best reference genes across different developmental stages, *RPL32* and *EF-1α* were recommended for tissue expression analysis, *RPL32* and *TBP* were identified as the optimal candidates for the temperature experiment, and *RPL32* and *RPL19* were selected for the insecticide treatment.

## Quantitative analysis of candidate reference genes based on geNorm

The geNorm algorithm provided the optimal number of reference genes for credible normalization under a given experimental condition. For all tested treatments, our results indicated that all pairwise variations were under a predetermined threshold of 0.15 (Fig 3), suggesting that two reference genes were sufficient for the normalization of the target gene. Taking the

**Table 2. Stability of reference gene expression under different experimental conditions calculated by four different analytical tools.**

| Experimental conditions | Reference genes | ΔCt | | Bestkeeper | | NormFinder | | geNorm | | Recommendation |
|---|---|---|---|---|---|---|---|---|---|---|
| | | Stability | Rank | Stability | Rank | Stability | Rank | Stability | Rank | |
| Development stage | *Actin1* | 0.628 | 3 | 0.337 | 2 | 0.259 | 4 | 0.528 | 5 | *TBP* |
| | *Actin2* | 0.706 | 6 | 0.343 | 3 | 0.364 | 6 | 0.554 | 6 | |
| | *EF-1α* | 0.641 | 4 | 0.230 | 1 | 0.306 | 5 | 0.420 | 3 | |
| | *GAPDH* | 0.970 | 8 | 0.845 | 8 | 0.608 | 8 | 0.700 | 8 | |
| | *α-TUB* | 0.837 | 7 | 0.469 | 6 | 0.511 | 7 | 0.611 | 7 | *RPL19* |
| | *RPL19* | 0.588 | 2 | 0.408 | 4 | 0.188 | 1 | 0.336 | 1 | |
| | *RPL32* | 0.647 | 5 | 0.536 | 7 | 0.257 | 3 | 0.498 | 4 | |
| | *TBP* | 0.585 | 1 | 0.432 | 5 | 0.191 | 2 | 0.336 | 1 | |
| Tissue | *Actin1* | 1.157 | 7 | 0.567 | 5 | 0.744 | 7 | 0.694 | 5 | *EF-1α* |
| | *Actin2* | 1.134 | 6 | 0.456 | 4 | 0.723 | 6 | 0.860 | 6 | |
| | *EF-1α* | 0.782 | 1 | 0.405 | 3 | 0.197 | 1 | 0.180 | 1 | |
| | *GAPDH* | 1.204 | 8 | 1.091 | 8 | 0.795 | 8 | 1.006 | 8 | |
| | *α-TUB* | 1.023 | 5 | 0.889 | 7 | 0.596 | 5 | 0.942 | 7 | *RPL32* |
| | *RPL19* | 0.930 | 3 | 0.091 | 1 | 0.348 | 3 | 0.600 | 4 | |
| | *RPL32* | 0.863 | 2 | 0.356 | 2 | 0.260 | 2 | 0.458 | 3 | |
| | *TBP* | 0.969 | 4 | 0.829 | 6 | 0.496 | 4 | 0.180 | 1 | |
| Temperature | *Actin1* | 0.904 | 6 | 0.983 | 6 | 0.450 | 6 | 0.603 | 5 | *RPL32* |
| | *Actin2* | 0.882 | 4 | 0.988 | 7 | 0.424 | 5 | 0.576 | 4 | |
| | *EF-1α* | 0.896 | 5 | 0.611 | 3 | 0.411 | 4 | 0.661 | 6 | |
| | *GAPDH* | 1.398 | 8 | 1.575 | 8 | 0.894 | 8 | 0.922 | 8 | *TBP* |
| | *α-TUB* | 0.795 | 3 | 0.664 | 5 | 0.303 | 3 | 0.464 | 3 | |
| | *RPL19* | 1.084 | 7 | 0.288 | 1 | 0.647 | 7 | 0.763 | 7 | |
| | *RPL32* | 0.692 | 1 | 0.601 | 2 | 0.022 | 1 | 0.354 | 1 | |
| | *TBP* | 0.722 | 2 | 0.655 | 4 | 0.137 | 2 | 0.354 | 1 | |
| Insecticide | *Actin1* | 1.479 | 8 | 1.177 | 7 | 0.939 | 8 | 1.034 | 8 | *RPL19* |
| | *Actin2* | 0.996 | 5 | 0.812 | 6 | 0.445 | 4 | 0.767 | 6 | |
| | *EF-1α* | 0.990 | 4 | 0.362 | 2 | 0.532 | 6 | 0.394 | 1 | |
| | *GAPDH* | 1.154 | 7 | 1.181 | 8 | 0.615 | 7 | 0.886 | 7 | |
| | *α-TUB* | 0.885 | 3 | 0.691 | 5 | 0.314 | 2 | 0.700 | 5 | *RPL32* |
| | *RPL19* | 0.876 | 1 | 0.482 | 3 | 0.359 | 3 | 0.394 | 1 | |
| | *RPL32* | 0.884 | 2 | 0.535 | 4 | 0.281 | 1 | 0.655 | 4 | |
| | *TBP* | 1.009 | 6 | 0.331 | 1 | 0.510 | 5 | 0.591 | 3 | |

developmental expression as an example, the first V-value < 0.15 was observed at V2/3, addressing that two reference genes were sufficient for reliable normalization.

## Validation of selected reference genes

To confirm the reliability of the selected reference genes, the expression of *sHSP20.0*, a target gene, was detected under different thermal treatments (Fig 4). In the temperature treatment (from 25 to 40 ˚C), when the optimal reference set (*RPL32* and *TBP*) or the optimal reference (*RPL32*) alone was utilized to normalize the target, the expressions of *sHSP20.0* were up-regulated by 3.4 and 2.6 times, respectively, and significant difference was detected ($P < 0.01$). In comparison, when the least stable reference gene *GAPDH* served as a normalizer, only a 0.8-fold increase was observed in *sHSP20.0* expression ($P < 0.05$), showing a decreased percentage of increase compared to those using the former two normalizers.

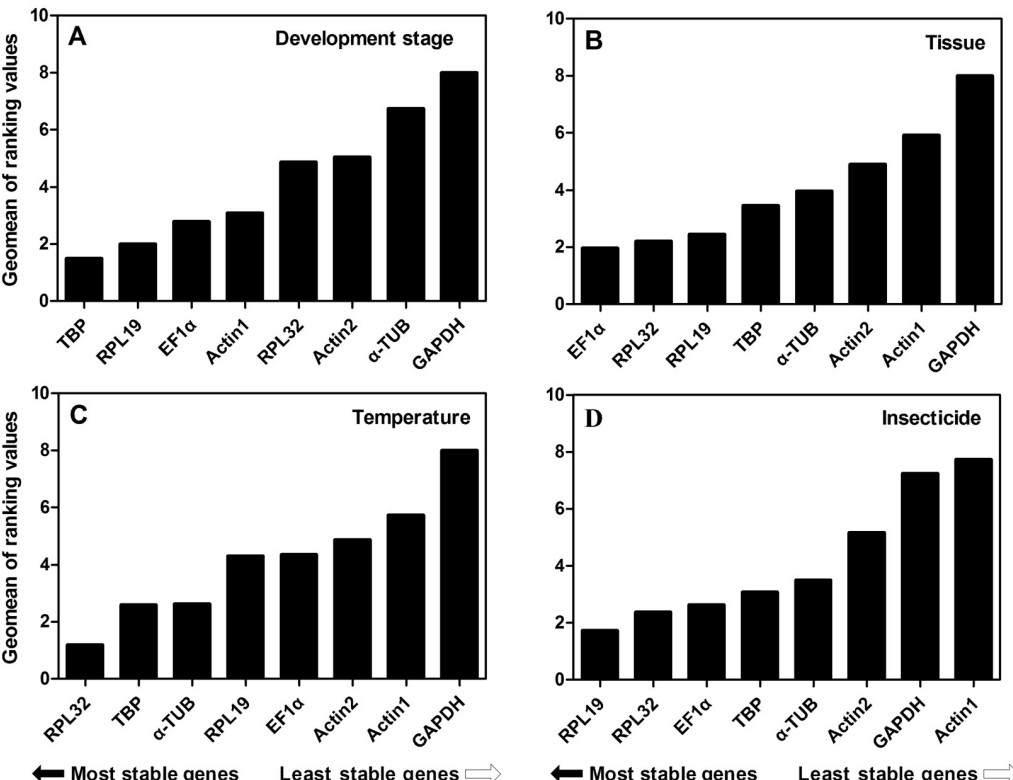

**Fig 2. The stability of candidate reference gene expression under different conditions based on RefFinder.** The expression stabilities of candidate reference genes were evaluated in diverse conditions including developmental stage (A), tissue (B), temperature stress (C), and insecticide exposure (D).

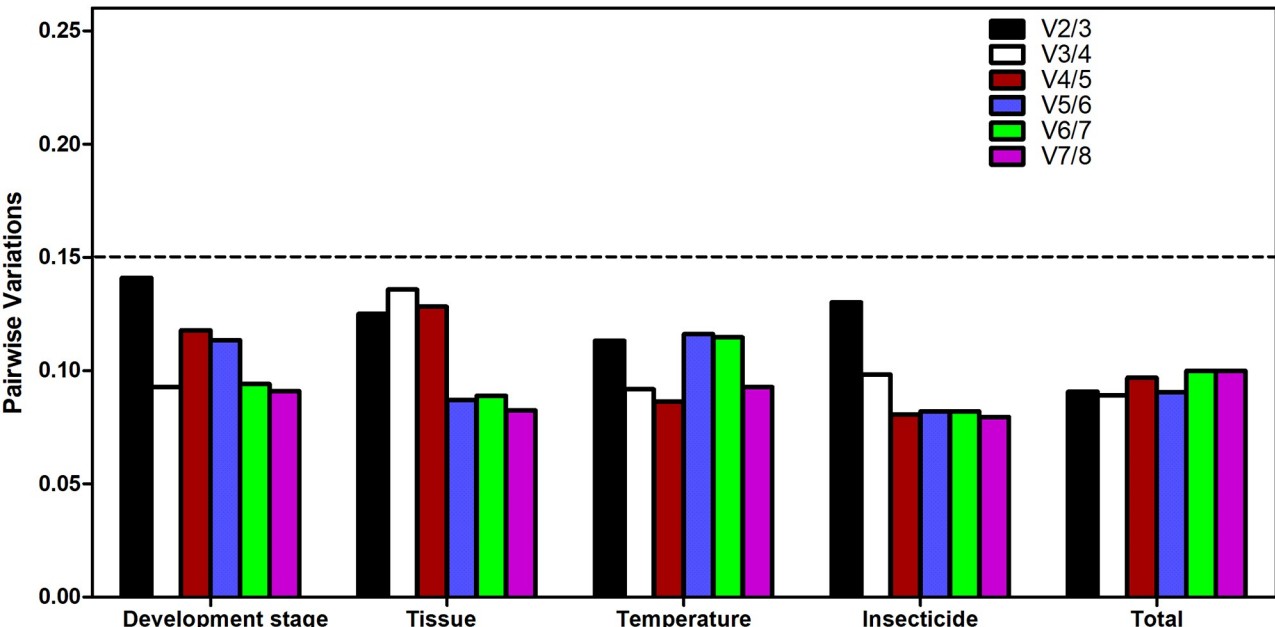

**Fig 3. Optimal number of reference genes used for normalization of gene expression by geNorm program.** A value of pairwise variation (Vn/Vn+1) below 0.15 suggested that no extra gene was required for normalization of gene expression in this condition.

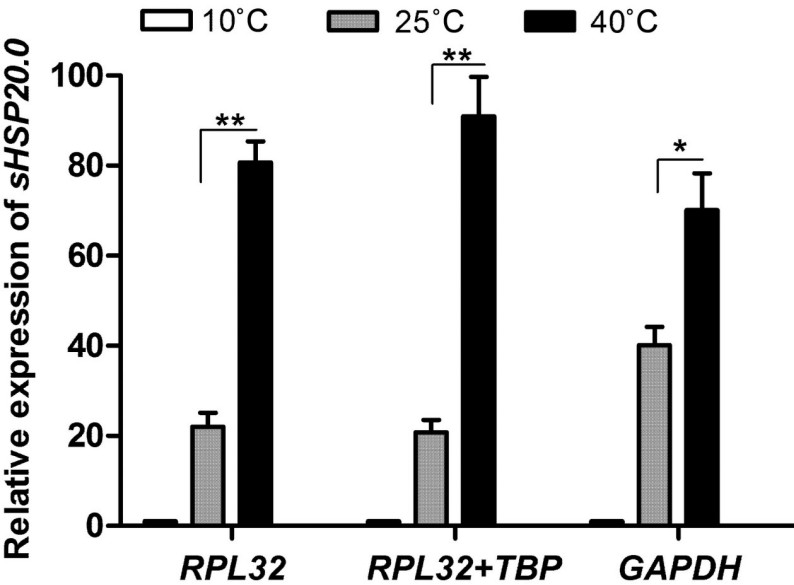

**Fig 4. The relative expression of target *sHSP20.0* normalized by the recommended set of references (*RPL32* and *TBP*), the optimal reference (*RPL32*) and the least stable gene (*GAPDH*), respectively.** Columns represent the expressions of *sHSP20.0* in *Phaedon brassicae* when subjected to cold shock (10˚C), heat shock (40˚C) and the control (25˚C) for 2 h. Error bars indicate SE; *, $P < 0.05$; **, $P < 0.01$ (Student's *t*-test by SAS 9.20).

## Discussion

Currently, RT-qPCR is extensively used to quantify gene expression [9]. Several concerns have been raised regarding the variations in RT-qPCR method, and normalization of RT-qPCR data with reference genes is a commonly used strategy to calibrate experimental errors introduced by RT-qPCR [10]. It is noteworthy that the selection of reference genes with low expression variation is a prerequisite to ensure valid normalization and avoid inaccurate quantification of gene expression [14, 19]. Previous studies have documented that many frequently used reference genes differed significantly in their expression profiles under different treatments [11, 19, 26]. Therefore, before RT-qPCR operation, each candidate reference gene should be evaluated under specific experimental conditions to ensure a stable expression level, and this step has become a routine practice before using them to normalize target gene expression.

Our results revealed that the stability of a reference gene could be changed under different experimental conditions. However, many previous studies utilized a single endogenous control for different treatments and life stages to quantify gene expression, which can significantly affect statistical analyses and may result in false data interpretation [27]. Therefore, it is imperative to identify the optimal reference genes for specific conditions in a given species. Furthermore, our study revealed that the stability ranking of these reference genes was variable in certain circumstances due to the different algorithms used in the four analytical tools. For instance, when *P. brassicae* was subjected to pesticide exposure, *RPL32* was listed as the optimal by NormFinder, and *TBP* was the top choice by Bestkeeper, whereas *RPL19* was recommended by geNorm. To address this challenge, an integrated analysis to evaluate the dataset becomes necessary, and adopting the multiple instead of a single normalizer for RT-qPCR analysis is in demand. For these purposes, RefFinder integrates the results of these computational programs and calculates a comprehensive ranking value for candidate genes [15, 19, 20], and meanwhile, a suite of reference genes is specifically recommended instead of a single

normalizer. Herein, the optimal number of reference genes was determined by geNorm, and our results proved that two references were sufficient for reliable normalization in each given condition.

Ribosomal proteins are known to play an essential role in ribosome assembly, and they, in conjunction with four ribosomal RNA (rRNA), make up the ribosomal subunits responsible for cellular protein translation [28]. Our results demonstrated that the ribosomal protein *RPL19* was expressed stably across life stages and under pesticide application, and *RPL32* showed high stability under thermal and pesticide treatment. Consistent with our results, many ribosomal proteins have been documented as the optimal reference for many insect species. Likewise, in the cabbage beetle *C. bowringi*, *RPL19* was identified as the optimal reference for different sexes and under photoperiod treatments [20]. In other coleopterans, *RP18* and *RP4* were regarded as the most stable house-keeping genes in *Leptinotarsa decemlineata* (Say) [29], *RpS9* showed a steady expression under a range of temperatures in *Diabrotica undecimpunctata howardi* [30], and *RPL22e* was selected as the suitable reference in different sexes of *Mylabris cichorii* [31]. Besides, *RpS3* and *RpS13A* showed the highest stability under ultraviolet irradiation in *Tribolium castaneum* [32]. Moreover, in the aphid parasitoid wasp *A. gifuensis*, *RPL13* was recommended as the optimal under diverse conditions including different developmental stages, sexes, and diverse diets [18]. In a tetranychid mite *Tetranychus urticae*, *Rp49* was suitable not only for host plant shift studies but also for the investigations of acaricide susceptible and resistant populations [33]. *EF-1α* is a ubiquitous and conserved cytosolic protein among eukaryotic organisms and is responsible for catalyzing the binding of aminoacyl-transfer RNAs to the ribosome [34, 35]. Our study indicated that *EF-1α* showed the best performance in diverse developmental stages and tissues. Studies of other coleopterans (i.e. *H. convergens* and *D. undecimpunctata howardi*) also showed that *EF-1α* acted as the best reference gene across life stages [15, 30]. Similar results were documented in several lepidopteran species (i.e. *M. separata*, *Danaus plexippus*, and *Diaphania caesalis*), where *EF-1α* was identified as the most stable reference across life stages and in tissues [12, 36, 37].

One surprising finding was that the traditional reference gene *GAPDH* was listed as the least reliable reference gene in most experimental conditions. Likewise, the instability of *GAPDH* expression has been documented in different developmental stages and tissues of *C. bowringi* [20] and *D. caesalis* [36], in *H. convergens* under thermal stress [15], and in *M. separata* after pesticide exposure [12]. Previous literature has documented that *GAPDH* functions as a glycolytic enzyme involved in glycolysis [38], and it was presumed that any perturbation toward energy metabolism would have a potential impact on *GAPDH* expression. Recent evidence suggests that *GAPDH* is associated with cell proliferation under adverse conditions where its catalytic activity is impaired [38]. Considering these issues, it is inappropriate to adopt *GAPDH* as a reference under several abiotic stress conditions, such as starvation, pesticides, and thermal stress.

To further validate the reliability of the optimal reference genes in *P. brassicae*, the expression of a target gene *sHSP20.0* was investigated under different thermal stress. As molecular chaperones, sHSPs assist in the correct folding of nascent proteins and combat protein aggregation induced by stresses, especially under thermal stress [24, 39]. In *T. castaneum*, *hsp18.3* was dramatically up-regulated in response to enhanced heat stress but not to cold stress [24]. Similarly, *sHSP19.1*, from the oak silkworm, was strongly induced after heat shock [40]. Our results showed that *sHSP20.0* expression was inconsistent when normalized to the least stable reference compared with that when normalized to the optimal reference set or the optimal reference alone. These findings revealed that the arbitrary selection of reference genes would lead to inaccurate or contradictory results for target genes [15, 41], and our results demonstrated

that the combined use of optimal reference genes ensures greater accuracy in gene expression analysis.

In conclusion, our results demonstrated that unstable reference genes might result in incorrect interpretation of RT-qPCR results, and the optimal reference gene recommendations could avoid such bias in normalization. To date, this is the first study to investigate candidate reference genes for gene expression analysis in *P. brassicae*, and our findings would lay a foundation for functional research in *P. brassicae*.

## Supporting information

**S1 Fig. The agarose gel electrophoresis of eight candidate reference genes.** M, marker. Templates in the PCR reactions were as follows: 1) *Actin2*, 2) *GAPDH*, 3) *RPL32*, 4) *α-TUB*, 5) *Actin1*, 6) *Ef-1α*, 7) *TBP*, and 8) *RPL19*.
(TIF)

**S2 Fig. Melting curve analysis of eight candidate reference genes.** The gene-specific amplification was confirmed by a single peak in melting-curve analysis.
(TIF)

**S3 Fig. Standard curves of eight candidate reference genes.**
(TIF)

**S1 Table. Sequences of candidate reference genes.**
(DOCX)

**S1 Raw images. The raw image of S1 Fig.**
(PDF)

## Author Contributions

**Conceptualization:** Ting Jiang, Xiangya Liu.

**Formal analysis:** Wanna Zhang.

**Funding acquisition:** Haijun Xiao.

**Methodology:** Long Ma, Ting Jiang, Xiangya Liu, Wanna Zhang.

**Resources:** Haijun Xiao.

**Validation:** Yingchuan Peng.

**Writing – original draft:** Long Ma.

**Writing – review & editing:** Long Ma.

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
