## [Decision Letter · Decision Letter 0]

21 Dec 2020

PONE-D-20-35478

Evalution of candidate reference genes for gene expression analysis ina Brassica Leaf Beetle, Phaedon brassicae (Coleoptera: Chrysomelidae)

PLOS ONE

Dear Dr. Zhang,

Thank you for submitting your manuscript to PLOS ONE. After careful consideration, we feel that it has merit but does not fully meet PLOS ONE’s publication criteria as it currently stands. Therefore, we invite you to submit a revised version of the manuscript that addresses the points raised during the review process. 

In particular, you should consider comments regarding English editing/usage, expansion of the Introduction to provide biological/physiological context for the study, and providing reasons for the methodological conditions (ie various pesticide concentrations) selected. In addition, please provide all corresponding accession numbers for the genes selected as well as transcriptomic datasets.  

We look forward to receiving your revised manuscript.

Kind regards,

J Joe Hull, Ph.D.

Academic Editor

PLOS ONE

Journal Requirements:

https://journals.plos.org/plosone/article?id=10.1371%2Fjournal.pone.0125868

In your revision ensure you cite all your sources (including your own works), and quote or rephrase any duplicated text outside the methods section. Further consideration is dependent on these concerns being addressed

4. Please upload a copy of Supporting Information Figures S1 and S2 which you refer to in your text.

Reviewers' comments:

Reviewer's Responses to Questions

**Comments to the Author**

1. Is the manuscript technically sound, and do the data support the conclusions?

Reviewer #1: Yes

Reviewer #2: Yes

Reviewer #3: Yes

2. Has the statistical analysis been performed appropriately and rigorously? 

Reviewer #1: Yes

Reviewer #2: Yes

Reviewer #3: Yes

3. Have the authors made all data underlying the findings in their manuscript fully available?

Reviewer #1: Yes

Reviewer #2: Yes

Reviewer #3: Yes

4. Is the manuscript presented in an intelligible fashion and written in standard English?

Reviewer #1: Yes

Reviewer #2: No

Reviewer #3: Yes

5. Review Comments to the Author

Reviewer #1: Ma et al. 2020 reports the validation of several housekeeping genes under different conditions in an important vegetable pest. This is one of several studies that document the importance of validating reference genes in different conditions, in order to obtain a set of standard genes that work best for appropriate conditions. The ms is well written, and analyses seem appropriate.

I have just a few comments below.

line 57: ‘remains’ is misspelled

line 97: I recommend to mention in this paragraph the 8 genes used in the study

line 115: …25°C ± 1, …

line 116: ..70% ± 10,…

line 126-129: how were the insects exposed, by feeding, filter paper petri dish, injection? Please clarify..

line 131: …2h and 4h…

line 131: …It was worth emphasizing that… this sentence is not needed, stat at ..Each experiment contained….

line 141: ‘specimen’ is misspelled

line 159: volume

line 214: ‘with’ regard is misspelled

line 290: Ribosomal proteins are known….

line 290-310: other example of chrysomelid species, Diabrotica undecimpunctata howardi, which is an important maize pest in North America, could be included in the discussion; RPS9 and EF-1α were documented each to be the most stable genes in one of the different conditions (Basu et al. 2019, https://doi.org/10.1038/s41598-019-47020-y).

line 335: delete ‘that’

line 340-345: I recommend to mention the best reference genes for each experimental conditions here.

Reviewer #2: Authors Ma et al. performed a study to evaluate candidate reference genes for molecular studies in Brassica Leaf Beetle, a serious pest of brassicaceous vegetables in East and South Asia. The research is important as identifying the most stable reference genes for a species is required for many molecular research based on the quantitative real-time PCR technique. In general, the experimental design and data analysis of this study are scientifically sound. However, the clarification of data presentation, language quality, discussion and citation need to be enhanced significantly. Following are some of my specific suggestions.

1. The English of this manuscript should be polished considerably. There are many errors (spelling, grammar, etc.) and clarification problems. Authors should rewrite some sentences to make it clear. Honestly speaking, asking for editorial help from a native speaker will be optimal.

1.1. Title: Evaluation, not Evalution

1.2. Title: change “in a Brassica Leaf Beetle” to “in the Brassica Leaf Beetle”

1.3. Line 55: change “insecticide” to “insecticides”

1.4. Rewrite the sentence in Lines 54-57. Two sentences instead of one will be ideal. Using However instead of but

1.5. Line 57: change “remians ambiguous” to “remain unknown”.

1.6. Line 57-58: rewrite this sentence. It is not clear! You did not mention anything of the functional gene research in your previous sentences.

1.7. Line 74: change “between” to “among”

1.8. Line 82: change “treatments” to “abiotic conditions”

1.9. Lines 87-91: using at least two sentences here instead of only one sentence

1.10. Line 119: low-case the subtitle except the first letter in order to keep consistence with other subtitles “Experimental Treatment and Sample collection”

1.11. Line 214: change “Wih” to “With”

1.12. Line 221: change “besides” to “and”

1.13. Line 230: add “ones” after “RPL19 as the most stable”

1.14. Lines 225-226: change this sentence to “ranked RPL32, TBP and α-TUB as the top three reference genes”

1.15. Line 227: add “the” – in “the” pesticide exposure experiment

2. Material and method

2.1. Line 111- please provide the background of pesticide usage of this insect population. Is there any pesticide application before and after keeping in the laboratory condition?

2.2. Line 143- please provide the accession numbers of these candidate reference genes in NCBI database

2.3. Line 146: any accession codes available for the RNA-seq transcriptome database of Phaedon brassicae? Or published papers? If yes, please cite them or provide this information

2.4. Line 165: Besides four individual programs you listed here, which program was used for ranking these candidate genes in general? (Table 2) RefFinder program? Please find and cite these references: Xie et al. Plant Molecular Biology 2012; Morales et al. International Journal of Biological Sciences 2016

2.5. Line 181: Please provide the accession number for sHSP20.0

3. Results:

3.1. Table 1. Please provide the accession numbers for these candidate genes

3.2. Figure 1: should be moved to supplementary data

3.3. Table 2: shows a “recommendation” of the most stable genes. However, in the M&M authors failed to provide the program used to recommend. Please reference previous published papers such as Xie et al. 2012; Morales et al. 2016, add more details in M&M and References

3.4. Figure 2: keep either plural or single consistent. For example, developmental stages, tissues, temperature conditions…

4. Citation and Introduction/Discussion:

There are many similar types of research published during past decade. However, authors only cited a few of them.

When author gave the Introduction, only 3 Phaedon brassicae papers were cited in the first paragraph (Lines 44-58)!! Some physiology background of this species is required in guiding authors to choose developmental stages, tissues, thermal stresses and insecticide stresses for their study.

Similarly, in the Discussion section, please cite more papers, especially some review or representative ones (e.g. insect pests, beneficial insects, mites, other arthropods or animals, plants, etc.) that have summarized the key questions in this research topic. Otherwise, this research is just simply mimicking previous publications in a different species!

Reviewer #3: The authors evaluated the potential of 8 genes as the reference gene that could be used in RT-qPCR assay. Four algorithms were applied to score each candidate gene across four types of samples and yielded 4 suites of optimal reference genes combination for each of the detected 4 given type of samples. The experiments were carefully designed and the results are reliable. This work is important to support the future gene functional study of Phaedon brassicae. The ms was carefully written with a clear logic flow. However, I would advise authors to get this manuscript checked by native English speaker. I believe this manuscript is a good piece of work but also has lot of scope for English language corrections. No major problem could be found. But some minor revisions need to be made before publication.

1. Why do you select the doses of acetamiprid at 2 μg/ml, dinotefuran at 5 μg/ml, and abamectin at 0.18 μg/ml? How did the authors obtain the sublethal dose of different pesticides? Why use PBS as control?

2. The discussion section need more concise.

Specific points:

1. Please add the version and literature for each software.

2. The gene sequences should be added in supplementary data.

3. The gene names should be italicized in entire MS.

4. The title needs correction. It should be..“ Evaluation of candidate reference genes for gene expression analysis in brassica leaf beetle, Phaedon brassicae (Coleoptera: Chrysomelidae)”.

5. line 117, “required” modified as “sufficient”.

6. Line 219: delete the blank space after “of”.

7. Line 336: delete “that” to correct the sentence.

8. Use RT-qPCR instead of qRT-PCR throughout the whole ms.

6. PLOS authors have the option to publish the peer review history of their article (what does this mean?). If published, this will include your full peer review and any attached files.

Reviewer #1: No

Reviewer #2: No

Reviewer #3: No

---

## [Author Response · Author response to Decision Letter 0]

22 Feb 2021

Dear editor and reviewers,

On behalf of my co-authors, we are grateful for giving us an opportunity to revise our manuscript, and we appreciate the anonymous reviewers for their positive and constructive comments on our manuscript. Based on these comments, we tried our best to polish the manuscript. The manuscript was modified systemically using the track change mode. The corrections are listed below point by point.

---

## [Decision Letter · Decision Letter 1]

6 Apr 2021

PONE-D-20-35478R1

Evaluation of candidate reference genes for gene expression analysis in the Brassica Leaf Beetle, Phaedon brassicae (Coleoptera: Chrysomelidae)

PLOS ONE

Dear Dr. Zhang,

Thank you for for the revised manuscript. After careful consideration, we feel that it has merit but does not fully meet PLOS ONE’s publication criteria as it currently stands. Therefore, we invite you to submit a revised version of the manuscript that addresses the points raised during the review process.

Your consideration for the Reviewers comments have strengthened the paper. Although the Reviewers for the most part are satisfied with the edits, because PLOS ONE does not utilize a copy editor I ask that you have the paper looked over again preferably by a native English speaker and/or a scientific editing service. Also, Reviewer 1 had a few minor suggestions (see below) that you might consider.  

We look forward to receiving your revised manuscript.

Kind regards,

J Joe Hull, Ph.D.

Academic Editor

PLOS ONE

Journal Requirements:

Reviewers' comments:

Reviewer's Responses to Questions

**Comments to the Author**

1. If the authors have adequately addressed your comments raised in a previous round of review and you feel that this manuscript is now acceptable for publication, you may indicate that here to bypass the “Comments to the Author” section, enter your conflict of interest statement in the “Confidential to Editor” section, and submit your "Accept" recommendation.

Reviewer #1: All comments have been addressed

Reviewer #2: All comments have been addressed

Reviewer #3: All comments have been addressed

2. Is the manuscript technically sound, and do the data support the conclusions?

Reviewer #1: Yes

Reviewer #2: Yes

Reviewer #3: Yes

3. Has the statistical analysis been performed appropriately and rigorously? 

Reviewer #1: Yes

Reviewer #2: Yes

Reviewer #3: Yes

4. Have the authors made all data underlying the findings in their manuscript fully available?

Reviewer #1: Yes

Reviewer #2: Yes

Reviewer #3: Yes

5. Is the manuscript presented in an intelligible fashion and written in standard English?

Reviewer #1: Yes

Reviewer #2: Yes

Reviewer #3: Yes

6. Review Comments to the Author

Reviewer #1: The authors made significant changes that improved the manuscript. I have a few minor changes below:

Line 63: do you mean 'LIFE' history.?

line 81: efficiency is misspelled

line 82: ...can AFFECT....

line 85: GENES

line 88: and TO eliminate...

line 96: delete 'speaking'

line 133: ....our results would facilitate future RESEARCH on ....

line 141: do you mean Petri dishes?

line 223: combination of reference GENES....

line 226: change 'OPERATED' by 'PERFORMED'

line 232: RANGING from ...

line 345: circumstances

line 374: howardi is part of the scientific name ...Diabrotica undecimpunctata howardi...

line 389: ..undecimpunctata howardi...

line 426: change 'give rise' to LEAD

Reviewer #2: Authors did a great job to revise their manuscript. All questions raised have been answered. I don't have any further questions.

Reviewer #3: (No Response)

7. PLOS authors have the option to publish the peer review history of their article (what does this mean?). If published, this will include your full peer review and any attached files.

Reviewer #1: No

Reviewer #2: No

Reviewer #3: **Yes: **HUIPENG PAN

---

## [Author Response · Author response to Decision Letter 1]

15 Apr 2021

Respond: Thanks for your constructive suggestion, and we have checked our writing thoroughly to prevent the misleading expression in the present version.

---

## [Editor Report · Decision Letter 2]

21 Apr 2021

PONE-D-20-35478R2

Evaluation of candidate reference genes for gene expression analysis in the Brassica Leaf Beetle, Phaedon brassicae (Coleoptera: Chrysomelidae)

PLOS ONE

Dear Dr. Zhang,

Thank you for submitting your manuscript to PLOS ONE. Unfortunately, the manuscript still needs extensive copy editing to rectify issues with sentence structure and verb tense. Frankly, in its present form the manuscript places too much of a burden on the reader to determine the authors' intended meaning for a number of sentences. I again suggest that you seek the input of a native English speaker or an English editing service.

We look forward to receiving your revised manuscript.

Kind regards,

J Joe Hull, Ph.D.

Academic Editor

PLOS ONE
---

## [Author Response · Author response to Decision Letter 2]

3 May 2021

Dear reviewers,

On behalf of my co-authors, we are grateful for giving us an opportunity to revise our manuscript, and we appreciate the anonymous reviewers for their positive and constructive comments on our manuscript. Based on these comments, we tried our best to polish the manuscript.

---

## [Editor Report · Decision Letter 3]

6 May 2021

Evaluation of candidate reference genes for gene expression analysis in the Brassica Leaf Beetle, Phaedon brassicae (Coleoptera: Chrysomelidae)

PONE-D-20-35478R3

Dear Dr. Zhang,

We’re pleased to inform you that your manuscript has been judged scientifically suitable for publication and will be formally accepted for publication once it meets all outstanding technical requirements.

Kind regards,

J Joe Hull, Ph.D.

Academic Editor

PLOS ONE
---

## [Editor Report · Acceptance letter]

21 May 2021

PONE-D-20-35478R3 

Evaluation of candidate reference genes for gene expression analysis in the Brassica Leaf Beetle, *Phaedon brassicae* (Coleoptera: Chrysomelidae) 

Dear Dr. Zhang:

I'm pleased to inform you that your manuscript has been deemed suitable for publication in PLOS ONE. Congratulations! Your manuscript is now with our production department. 

Kind regards, 

on behalf of

Dr. J Joe Hull 

Academic Editor

PLOS ONE